# The cost effectiveness of early assessment and intervention by a dedicated health and social care professional team for older adults in the emergency department compared to treatment-as-usual: Economic evaluation of the OPTI-MEND trial

**Dominic Trépel**[1,2,3☯]*, **Manuel Ruiz-Adame**[1,4☯], **Marica Cassarino**[5], **Elayne Ahern**[1,6], **Collette Devlin**[7], **Katie Robinson**[7], **Íde O'Shaughnessy**[7], **Gerard McCarthy**[5], **Cian Corcoran**[1], **Rose Galvin**[7]

1 Trinity College Dublin, Dublin, Ireland, 2 Global Brain Health Institute, Trinity College Dublin, Dublin, Ireland, 3 University California San Francisco, San Francisco, California, United States of America, 4 Applied Economics Department, Department Campus of Melilla, University of Granada, Melilla, Spain, 5 School of Applied Psychology, University College Cork, Cork, Ireland, 6 Department of Psychology, University of Limerick, Limerick, Ireland, 7 Health Research Institute, School of Allied Health, Ageing Research Centre, University of Limerick, Limerick, Ireland

☯ DT and MR-A are Joint first authors
* trepeld@tcd.ie

## Abstract

### Background

Over 65s are frequent attenders to the Emergency Department (ED) and more than half are admitted for overnight stays. Early assessment and intervention by a dedicated ED-based Health and Social Care Professionals (HSCP) team reduces ED length of stay and the risk of hospital admissions among older adults while improving patient health-related quality-of-life and satisfaction with care. This study aims to evaluate whether augmenting the treatment as usual for older adults admitted to ED is cost-effective.

### Methods and findings

Cost-effectiveness analysis (CEA), conducted alongside the OPTI-MEND randomised controlled trial of 353 patients aged ≥65 with lower urgency complaints compared the effectiveness of early assessment and intervention by a dedicated HSCP team in the ED to treatment as usual (TAU). An economic analysis estimated the average cost per older adults randomised to the HSCP team, and compared to TAU, how contact with HSCP team changed health care use, and associated total costs, and estimated the effect of HSCP on Quality-Adjusted Life Years (QALYs). Within the OPTI-MEND trial, the average cost of a contact with the HSCP team during ED attendance is estimated to be €801 per patient. Compared to TAU, the incremental QALY of intervention is 0.053 (95% CI: 0.023 to 0.0826,

**Data Availability Statement:** Yes - all data are fully available without restriction. All data files are available within the paper or https://osf.io/5wpjm/.

**Funding:** Funded: Authors' Initials: KR, GM, RG. Grant Number: RCQPS-2017-2 Full Name: Health Research Board of Ireland URL: https://www.hrb.ie The funders had no role in study design, data collection and analysis, decision to publish, or preparation of the manuscript.

**Competing interests:** The authors have declared that no competing interests exist.

**Abbreviations:** ADL, Activities of daily living; CEA, Cost-effectiveness analysis; CEAC, Cost-effectiveness acceptability curve; CEP, Cost-effectiveness plane; CI, Confidence Interval; CONSORT, Consolidated standards of reporting trials; ED, Emergency department; GP, General practitioner; HCU, Healthcare use; HSCP, Health and Social Care Professional; HSE, Health Service Executive; ICER, Incremental cost-effectiveness ratio; LoS, Length of Stay; MTS, Manchester triage system; QALY, Quality-adjusted life year; TAU, Treatment as usual.

p<0.0001). Accounting for cost savings because of contact with HSCP team, the average incremental saving in the total cost, compared to TAU, is -€6,128 (95% CI: -€9,217 to -€3,038, p<0.0001). Given the incremental health gains and significant cost savings, boot-strapped cost CEA suggests that dedicated HSCP care dominates over TAU for low urgency older adults attending the ED.

## Conclusions

A dedicated HSCP team in the ED significantly improves overall health for lower acuity older adults and, by reducing inpatient length of stay, results in staggering cost savings. This economic evaluation conducted on the OPTI-MEND trial provides convincing evidence that HSCP should be adopted as part of treatment as usual in Irish EDs.

## Trial registration

ClinicalTrials.gov, NCT03739515; registered on 12th November 2018. https://classic.clinicaltrials.gov/ct2/show/NCT03739515.

## Introduction

Older adults are frequent users of emergency departments (EDs) [1,2] and, under treatment as usual (TAU), more than half of ED attendances result in inpatient admissions with a median length of stay of nine nights (interquartile range 5 to 24 nights) [3]. Hospital admissions are associated with increased ED wait times [4], shortages in hospital beds and complex admission pathways result in delayed length of stay (LoS) in the ED. The OPTI-MEND trial tested the effect of adding a dedicated team of Health and Social Care Professionals (HSCP) to the ED and concluded that early assessment and intervention for low urgency older people can facilitate shorter stays in the ED, reduced risk of hospital admissions and improve satisfaction with overall care. This health economic study extends on the clinical trial and, by using OPTI--MEND trial data, conducts a cost effectiveness analysis to inform policy on whether HSCPs may represent value for money.

Appraisal of the best available evidence suggests that interventions centred around care coordination in the ED may increase clinical effectiveness for older adults [5]. Care Coordination Teams in the ED reduce rates of hospital admission [6], which motivated the hypothesis that, allocating a dedicated HSCP team who conducts early assessment and intervention, would result in better clinical and economic outcomes for lower acuity older adults. Specifically, HSCP teams are demonstrated to result in favourable discharge outcomes to home and enhanced continuity of care in the community [7,8] which, on aggregate, should improve health-related quality-of-life and reduce demand for limited healthcare resources.

The OPTI-MEND trial was conducted to determine whether ED-based interdisciplinary HSCP teams are effective to reduce LoS in the ED and incidence of hospital admission among older adults [3]. The HSCP team allocated into the ED included one additional senior physiotherapist, one senior occupational therapist, and one senior medical social worker to provide early assessment and intervention to lower acuity older adults. The team conducted interdisciplinary assessments of functional and mobility status, cognition, and psychosocial needs and subsequent interventions were tailored to individual older adults' needs (including, but not limited to, patient and family education on the outcome of the HSCP assessment and ED

discharge plan, prescription of mobility aids and enabling Activities of Daily Living (ADL) equipment, provision of home exercise programmes, education of self-management strategies and onward referral to alternative care pathways). The primary clinical study found that, compared to usual ED care, HSCP teams were clinically effective in reducing ED length of stay (6.4 versus 12.1 median hours, $p < 0.001$), and incidence of hospital admissions (19.3% versus 55.9%, $p < 0.001$); this motivated the further hypothesis that dedicated HSCP teams for older adults should also be a cost effective service, and potentially cost saving, within the Irish health system.

To determine whether dedicated HSCP teams for older adults in the ED represent value to the Irish health system, the national guidance recommends that incremental health gains from intervention over TAU be expressed as *Quality-Adjusted Life Years* (QALYs), and all costs relevant to a health and social care budget should be considered [9]. To be considered cost effective, the incremental cost effectiveness ratio of HSCP plus usual ED care, compared to usual ED care alone, would need to demonstrate producing health gain for less than the cost-effectiveness threshold, which is currently set at €45,000 per QALY in Ireland.

Economic evaluations of ED models of care have been shown, through systematic searches, to be largely absent in the evidence base [10]. This paper reports cost-effectiveness analysis, conducted alongside the OPTI-MEND trial, with a view to informing Irish decision-makers on whether routinely allocating HSCP teams, that provide early assessment and intervention for low urgency older people, represents value for money, compared to TAU.

## Methods

All methods regarding the conduct of this economic evaluation conducted alongside the OPTI-MEND trial were described in a Health Economic Analysis Plan [11].

### Trial design

A single-centre, parallel group, randomised controlled trial was conducted in the ED of a regional university teaching hospital in the Mid-West of the Republic of Ireland. The trial was registered on ClinicalTrials.gov (NCT03739515) and a protocol detailing the clinical- and cost-effectiveness analyses were published in advance [12]. The study received ethics approval from the Health Service Executive (HSE) Mid-Western Regional Hospital Research Ethics Committee on 20th September 2018 (Ref: 103/18).

Participant inclusion followed specific eligibility criteria described in Table 1. Recruitment of participants took place between December 2018 and May 2019. After giving written consent to take part, each participant underwent a baseline assessment and were then randomly allocated to the intervention or control group. A total of 353 older people aged ≥65 years were randomised to either receive HSCP plus TAU (n = 176), or TAU (n = 177). Full details on patient inclusion criteria are described elsewhere.

The research nurse (CD) who conducted the evaluations was blind to group allocation. Only the research nurse and the HSCP team had access to information that could identify the participants during data collection. Once all data were collected, data were anonymised and the final dataset that was used for analysis contained no identifiable information.

### Intervention and control groups

To create the HSCP team, three full-time senior healthcare professionals were employed and allocated to work in the ED: one physiotherapist, one occupational therapist and one medical social worker, all at senior level. Participants who were eligible to be seen by the HSCP team were identified either through the ED triage system, or via consultation with the Emergency

**Table 1. Eligibility criteria of trial participants at enrolment.**

| Inclusion criteria | Exclusion criteria |
|---|---|
| Aged ≥65 years | Aged under 65 years |
| MTS 3–5 | MTS 1–2* |
| Off baseline mobility and functional status | Neither the patient nor the carer can communicate in English sufficiently to complete informed consent or baseline assessment |
| Capacity and willingness to provide informed consent | Lacking capacity to provide informed consent ** |
| Presenting during HSCP operational hours (8am-5pm Monday-Friday) | Presenting outside HSCP operational hours (5pm-8am or on Saturday/Sunday) |
| Presenting with any of the following complaints, as per Manchester Triage System [13]: *Before medical work-up*: Limb problems; Falls; Unwell adult; Back pain; Urinary problems, or Ear and facial problems | Presenting with complaints other than described in the inclusion list. |

MTS = Manchester Triage System.

* MTS score 1–2 only recruited after Emergency Medicine diagnostic work-up and suitability for HSCP assessment determined.

**In cases where there was a clinical concern regarding capacity to consent, the 4AT tool was used to screen for cognitive impairment and in participants where there was evidence of moderate-profound impairment, the patient's nominated contact person was contacted for consent.

Medicine staff. The control group received treatment as usual (TAU) chosen because it represented routine ED care that patients would ordinarily receive on attendance at the ED and allowed incremental cost effectiveness analysis.

## Cost required to provide HSCP team

To inform the cost of the intervention within the base-case cost-effectiveness analysis, the OPTI-MEND trial budget for allocating the HSCPs, over TAU, were utilised. The average cost per participant was calculated as the total budget divided by the number of trial recipients.

## Resource use and associated costs

In addition to considering the cost of providing a HSCP team in the ED, and determining how this compares to TAU, resource use data were collected from all participants by a trained research nurse blind to group allocation. Resource use data was gathered from the hospital database of service use following discharge and included number of visits (if any) to the General Practitioner (GP), public health nurse, home help/home support, private consultations, outpatient department visits, and allied health service use.

The economic evaluation sought to examine wider resource use, and the associated cost, both immediately following the ED index visit, and in the interval between the two successive trial follow-up timepoints (30-days and 6-months). Immediately following presentation to the ED, time from ED registration to discharge was measured in hours and, where participants were admitted as inpatients, their length of inpatient stay was captured. Participants were also followed up at 30 days after their index visit to ED and the number of unscheduled ED re-visit and, where applicable, length inpatient stay, were captured. The final follow-up was conducted approximately 6 months after the index visit and individuals resource use after the date of their 30-day follow up was captured and reported unscheduled ED re-visit, inpatient length of

stay, outpatient contact and community contacts (specifically with either general practitioner, nurse, physiotherapist, occupational therapist, dietician, or podiatrist).

For all items of resource use captured, a related Irish Unit Cost was identified (see Table 3) and, for each participant, the quantity of each resource item was multiplied by the related Irish Unit cost. Summing all individual costs, a *total cost* per participant was calculated and, by adding the incremental cost related to their intervention group (HSCP + TAU or TAU), a group average total cost, relevant from the perspective of the wider healthcare system, was calculated. Formally, the Total Cost equation is:

$$\begin{aligned} Total\ Cost = &\ (Cost\ of\ Intervention + ED\ length\ of\ stay + Inpatient\ length\ of\ stay)_{T0} \\ &+ (ED\ visits + Inpatient\ length\ of\ stay)_{T1} \\ &+ (ED\ visits + Inpatient\ length\ of\ stays + Outpatient\ contact + Community\ Contacts)_{T2} \end{aligned}$$

For each arm of the study, the average use of each resource, the associated average cost per item and the total average cost are summarised.

## Health outcomes for economic evaluation

Participants' responses to the EQ-5D-5L questionnaire were used to estimate health states utilities using the Irish value set [14]. Using an area-under-the-curve approach, the estimated health state utility at each timepoint and the specific dates of data collection, Quality-Adjusted Life Years (QALYs) were estimated across all timepoints.

## Cost-effectiveness analysis

Unadjusted Incremental Cost Effectiveness Ratio (ICER) for each intervention group are calculated compared to treatment as usual (TAU) using the following formula:

$$ICER = \frac{\overline{Cost}_{HSCP} - \overline{Cost}_{TAU}}{\overline{QALY}_{HSCP} - \overline{QALY}_{TAU}}$$

where $\overline{Cost}$ is the average total costs and $\overline{QALY}$ is the average effect, expressed in terms of Quality-Adjusted Life Years.

To account for the joint distributions of cost and QALYs, the differences between groups were estimated using Zellner (1962) Seemingly Unrelated Regression Equation (SUR) [15]; SUR is selected as it is considered more efficient over unrelated ordinary least suare regression and reports correlation between costs and effects [16,17]. Non-parametric bootstrapping (10,000 replication) was conducted on random samples of the observed data and the results of the bootstrap are presented as a scatter plot on the cost effectiveness plane. Furthermore, joint distribution of costs and outcomes were illustrated using 50%, 75% and 95% confidence ellipses surrounding the ICER, indicating on the CE plane, the probability space within which we are confident the true ICER is found.

Guidelines for the Economic Evaluation of Health Technologies in Ireland require that probability analysis present "*the probability of an ICER is being below €20,000 and €45,000 per QALY, respectively*" [9]. Where relevant, the probability of HSCP being cost-effective at these willingness-to-pay thresholds were calculated and how the probability of HSCP being cost effective increases as willingness-to-pay increases, a cost effectiveness acceptability curve (CEAC) is generated. No discounting was applied as the study was less than 12 months in duration.

## Patient and public involvement

The design and implementation of the trial was informed by extensive consultation with key ED stakeholders, including ED patients and caregivers, as well as hospital and ED medical, nursing and HSCP staff [18].

## Results

Inspection of baseline characteristics indicated that trial randomisation produced well balanced groups across most characteristics and that the trial was sufficiently powered for cost effectiveness analysis.

The Consolidated standards of reporting trials (CONSORT) diagram was adapted to report key variables and explain the sample available for complete case cost-effectiveness analysis (see Fig 1). Overall, OPTI-MEND randomised 353 participants to either HSCP + TAU (n = 177) or TAU (n = 176) and all participants completed the intervention. For complete case cost effectiveness analysis, missing responses to EQ-5D-5L and/or dates for this data or missing health care use data (HCU) resulted in omission from the final analysis. Attrition at either the 30-day or 6-month follow-up was documented as either lost to follow up or participants discontinued in the study. For participants who died during the study, where the date of death was obtained, they were reported as "deaths" between the timepoint and were included in the complete case analysis (i.e., their health utility and health care use being zero on the date of death onwards).

With reference to trial budgets, the HSCP team were employed for a period of six months as part of the OPTI-MEND trial study at a cost of €118,792.89 for the duration of the trial. A budget of €7,500 was allocated to cover cost of aids and appliances for participants during the intervention and a dedicated assessment room in the ED for €14,600 for the six-month period. This total required budget was €140,892.89.

To calculate individuals cost related to their health care use, Table 2 provides the unit costs use to convert resource usage into costs.

Table 3 reports, by treatment group, the average health care use (left) and associated cost (right) for all resource use items obtained in the trial.

Table 4 reports the average and 95% Confidence Intervals for each timepoint, unadjusted utilities and costs by timepoint, and across the whole study, QALY and total costs and between group difference of QALYs and Total Cost. The difference in QALYs and Total Cost were subject to bootstrapping (10,000 replication) to provide unbiased 95% confidence intervals and finds the unadjusted between-group difference in QALYs is 0.053 (bootstrapped 95% CI: 0.019 to 0.086) and in total cost -€6,128 (bootstrapped 95% CI: -€9,180 to -€3,075).

To model the joint distribution of costs and QALYs for incremental cost effectiveness analysis of HSCP, and to control for baseline utility, seemingly unrelated regression was performed on the n = 322 complete cases available across all time points. Regression of the joint distribution find that total costs and QALYs were significantly correlated, and treatment group explained a large proportion of the variance in QALY ($R^2$ = 0. 2442) and a smaller proportion of the variance in total cost ($R^2$ = .0448). Correlation between Total Cost and QALYs was -0.2803 and negative correlation indicates individuals with worse outcomes have higher costs. Accounting for correlation in the joint distribution, the dedicated HSCP intervention reduced total cost to healthcare by €6,128 (95% CI: -€9,217 to -€3,038, p<0.001) and resulted in an incremental QALY of 0.0529 (95% CI: 0.0231 to 0.0826) (see Table 5).

Fig 2 illustrates results from the analysis of the uncertainty of the joint distribution of total cost and QALY. As the majority of bootstrapped replicates call in the bottom right quadrant, this indicates that HSCP has a 99.85% certainty that HSCP + TAU *dominates* (i.e. is more

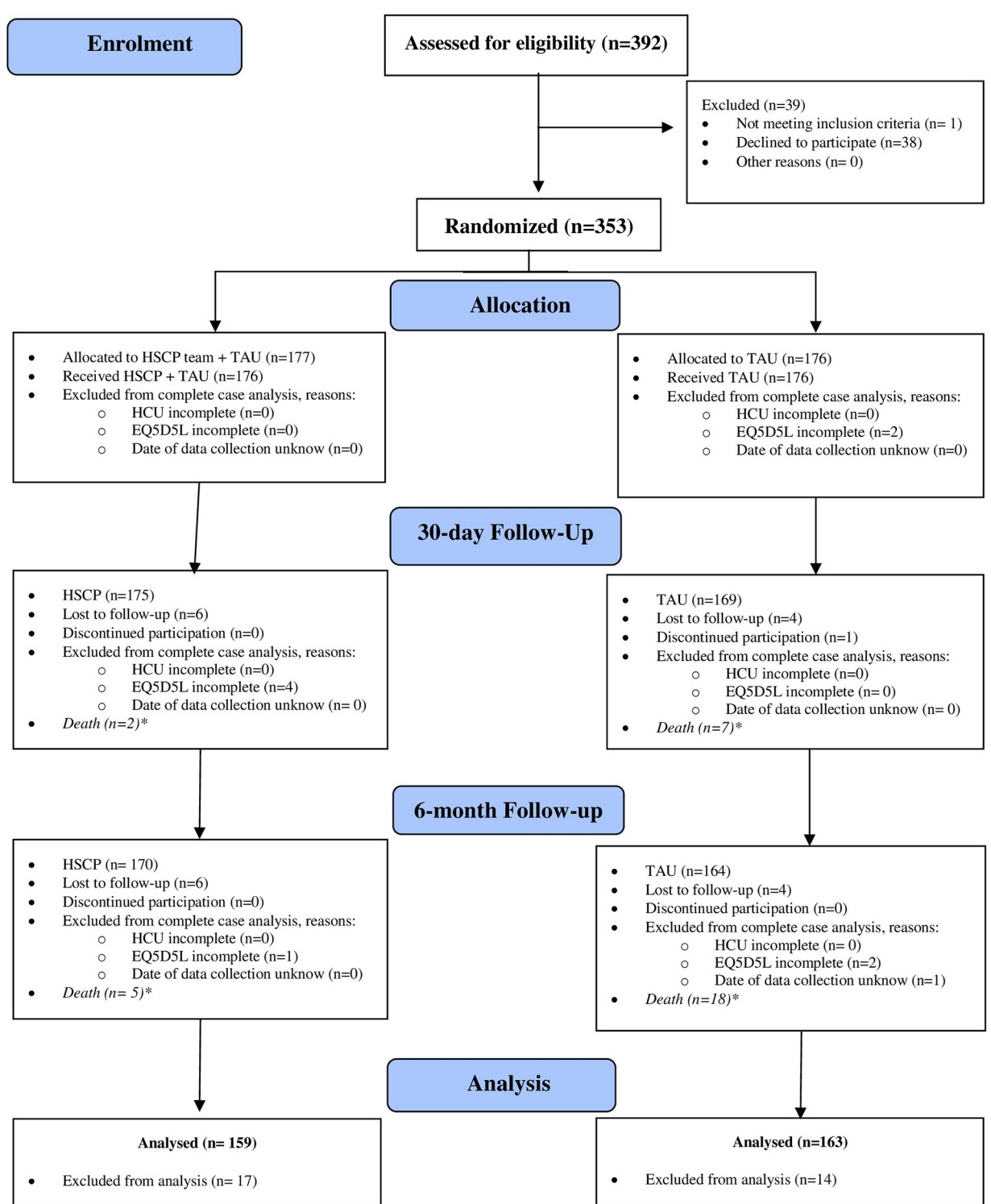

**Fig 1. CONSORT 2010 flow diagram.** In contrast the clinical effectiveness analysis [3], this diagram specifically explains data available for use in complete case cost effectiveness analysis.

**Table 2. Unit costs (in 2019 € prices).**

| Resource use items | Unit cost (€) | Sources |
|---|---|---|
| **HSCP intervention** | €800.53 | See Appendix 1 |
| **Cost associated with consequences of HSCP intervention** | | |
| Average cost of ED admission | €264.98 | See Appendix 2 |
| Cost per patient hour in ED | €14.63 | See Appendix 2 |
| Inpatient elective stay: (national average per night) | €933.00 | Gillespie (2022) [19] |
| Inpatient emergency stay (national average per night) | €933.00 | Gillespie (2022) [19] |
| Outpatient consultation (average cost) | €136.00 | Gillespie (2022) [19] |
| General Practitioner appointment | €60.00 | Gillespie (2022) [19] |
| Nurse | €56.00 | Smith (2021) [20] |
| Physiotherapist | €65.00 | Smith (2021) [20] |
| Occupational Therapist | €65.00 | Smith (2021) [20] |
| Dietician | €60.00 | Smith (2021) [20] |
| Podiatrist | €65.00 | Smith (2021) [20] |

effective and saves money) over TAU alone suggesting a high probability the intervention should replace the usual arrangement of care.

## Impact of patient and public involvement

Qualitative insights gathered during the consultation identify enablers and challenges associated with the introduction of the HSCP team and the trial data collection [15].

**Table 3. Unadjusted resource use, and associated costs, by treatment group (Source: Medical records).**

| Healthcare Resource items (by timepoint) | Resource use | | | | | | Associated costs | | | | | |
|---|---|---|---|---|---|---|---|---|---|---|---|---|
| | HSCP | | | TAU | | | HSCP | | | TAU | | |
| | Mean | Sd | n | Mean | sd | n | Mean (€) | sd (€) | n | Mean (€) | sd (€) | n |
| **Baseline:** | | | | | | | | | | | | |
| HSCP team intervention | 1 | 0 | 177 | 0 | 0 | 177 | 801 | 0 | 176 | 0 | 0 | 177 |
| ED length of stay (hours) | 11.502 | 12.729 | 176 | 18.113 | 19.414 | 177 | 168 | 186 | 176 | 265 | 284 | 177 |
| Hospital length of stay (days) | 2.119 | 6.068 | 176 | 9.322 | 15.677 | 177 | 1,977 | 5,662 | 176 | 8,697 | 14,626 | 177 |
| **30-day follow up:** | | | | | | | | | | | | |
| Number of unscheduled ED re-visit | 0.222 | 0.526 | 176 | 0.169 | 0.47 | 177 | 59 | 139 | 176 | 45 | 125 | 177 |
| Inpatient admission: length of stay (days) | 1.159 | 4.028 | 176 | 1.373 | 4.808 | 177 | 1,081 | 3,758 | 176 | 1,281 | 4,486 | 177 |
| **6-month follow up:** | | | | | | | | | | | | |
| Number of unscheduled ED re-visit | 0.301 | 0.571 | 176 | 0.367 | 0.704 | 177 | 80 | 151 | 176 | 97 | 186 | 177 |
| Inpatient admission 1: length of stay (days) | 1.812 | 7.145 | 176 | 2.068 | 5.848 | 177 | 1,691 | 6,666 | 176 | 1,929 | 5,456 | 177 |
| Inpatient admission 2: length of stay (days) | 0.415 | 2.741 | 176 | 0.531 | 2.518 | 177 | 387 | 2,557 | 176 | 495 | 2,349 | 177 |
| Inpatient admission 3: length of stay (days) | 0.193 | 2.563 | 176 | 0.102 | 1.353 | 177 | 180 | 2,391 | 176 | 95 | 1,262 | 177 |
| Outpatient appointments | 0.835 | 1.261 | 176 | 1.017 | 1.653 | 177 | 114 | 171 | 176 | 138 | 225 | 177 |
| Community contacts: General Practitioner | 1.864 | 2.41 | 176 | 1.565 | 2.288 | 177 | 112 | 145 | 176 | 94 | 137 | 177 |
| Community contacts: Nurse | 1.08 | 3.673 | 176 | 1.729 | 5.745 | 177 | 60 | 206 | 176 | 97 | 322 | 177 |
| Community contacts: Physiotherapist | 0.875 | 2.52 | 176 | 0.424 | 1.351 | 177 | 57 | 164 | 176 | 28 | 88 | 177 |
| Community contacts: Occupational Therapist | 0.097 | 0.333 | 176 | 0.141 | 0.619 | 177 | 6 | 22 | 176 | 9 | 40 | 177 |
| Community contacts: Dietician | 0 | 0 | 176 | 0.006 | 0.075 | 177 | 0 | 0 | 176 | 0 | 5 | 177 |
| Community contacts: Podiatrist | 0.091 | 0.457 | 176 | 0.056 | 0.409 | 177 | 6 | 30 | 176 | 4 | 27 | 177 |
| **Total cost** | - | - | - | - | - | - | **€6,779** | **€12,083** | **176** | **€13,275** | **€16,976** | **177** |

**Table 4. Mean (95% Confidence Intervals) for unadjusted utilities and costs (by timepoint), QALY and total costs (across all timepoints) and between group difference of QALYs and total cost.**

| Timepoint | HSCP | | TAU | | Between group difference |
|---|---|---|---|---|---|
| | Costs | Outcomes | Costs | Outcomes | |
| Baseline | €2,741 (€1,909 to €3,574) | 0.484 (0.428 to 0.54) | €8,203 (€6,171 to €10,235) | 0.484 (0.426 to 0.542) | - |
| 30-day | €1,170 (€563 to €1,777) | 0.7 (0.65 to 0.75) | €1,440 (€710 to €2,169) | 0.623 (0.561 to 0.684) | - |
| 6-months | €2,725 (€1,364 to €4,087) | 0.773 (0.73 to 0.817) | €3,122 (€1,995 to €4,249) | 0.701 (0.644 to 0.758) | - |
| QALY* | - | 0.344 (0.323 to 0.364) | - | 0.291 (0.264 to 0.317) | 0.053 (0.019 to 0.086) |
| Total cost* | €6,637 (€4,746 to €8,528) | - | €12,764 (€10,344 to €15,185) | - | -€6,128 (-€9,180 to -€3,075) |

* Bootstrapped 95% confidence intervals (10,000 replications).

**Table 5. Seemingly unrelated regression of cost and QALYs, controlling for baseline utilities (n = 322).**

| Variables | Total cost (HSE perspective, €, 95% C.I.) | QALY (95% CI) |
|---|---|---|
| Treatment | -€6,128 (-€9,217 to -€3,038) *** | 0.0529 (0.0231 to 0.0826) *** |
| Baseline utility | | 0.1821 (0.1431 to 0.2211) *** |
| Constant | €12,764 (€10,594 to €14,935) *** | 0.2026 (0.1745 to 0.2308) *** |

$R^2$(Total Cost): 0.0448; $R^2$(QALY): 0.2442; Correlation matrix of residuals of Total Cost and QALYs: -0.2803;
Breusch-Pagan test of independence: $\chi^2$(1): 25.303***.
Significance levels: ***: $p < 0.001$; **: $p < 0.005$.

## Discussion

The OPTI-MEND trial was performed on the hypothesis that, early assessment and intervention for low urgency older people can facilitate shorter stays in the ED, reduces risk of hospital admissions and improve satisfaction with overall care. Analysis of clinical effectiveness demonstrated such HSCP teams significantly reduce ED LoS, and incidence of hospital admissions [3] and, building upon these initial findings, this formal cost-effectiveness analysis now confirms the magnitude of potential cost savings the Irish health system, as well as significant improvement in health.

 The economic evaluation conducted alongside the OPTI-MEND trial firstly estimates that the average cost of a contact with the HSCP team during ED admission is €801. In line with HIQA guidance from the *Economic Evaluation of Health Technologies in Ireland* require that probability analysis present "*the probability of an ICER is being below €20,000 and €45,000 per QALY, respectively*" [9]. Because of contact with HSCP team, there is an average incremental saving in the total cost, compared to TAU, of -€6,128 per patient, largely driven by averting inpatient admission and stay. As effectiveness analysis show an average benefit of 0.053 additional per QALYs and given the treatment results in overall cost saving, there is certainty from OPTI-MEND data that, HSCP teams are cost effective and may in fact 'dominate' usual care (i.e. that is would be efficient use of resource to replace the current arrangement of care in the subpopulation).

 The OPTI-MEND trial has shown that a dedicated ED-based HSCP team, as compared to TAU, has positive clinical outcomes that allow a higher use of services for more populations (e.g. by reducing inpatient length of stay, lower rates of hospital admission) and a high reduction in cost per patient. From this analysis, we can reliably conclude that HSCP represents value to the Irish health system and should be adopted as part of treatment as usual in Irish

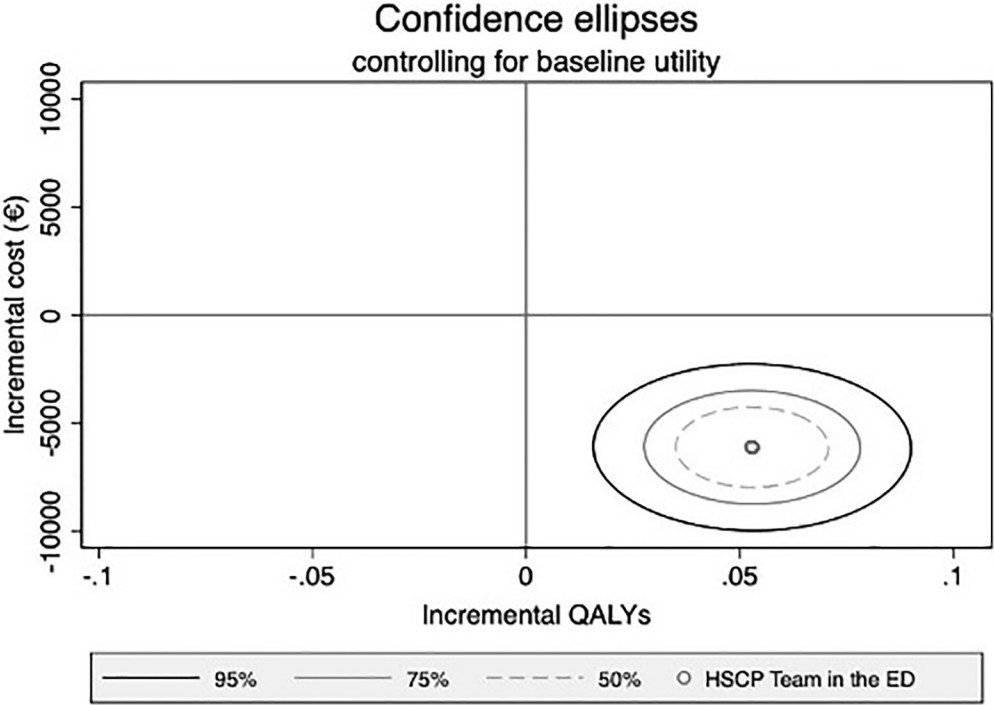

**Fig 2. Cost-effectiveness plane showing uncertainty in the joint distribution of cost and QALYs that surround the incremental cost-effectiveness ratio (ICER).**

EDs. While these dedicated HSCP teams are currently in situ across the majority of ED in Ireland, further work is ongoing to establish core standards of care across these teams.

The study was carried out on data collected in one setting within the Irish context and the results may not be generalisable to other contexts where healthcare infrastructure, processes and costs may be different.

## Supporting information

**S1 File. Supplementary materials (Appendix 1 and 2).**
(DOCX)

## Author Contributions

**Conceptualization:** Dominic Trépel, Manuel Ruiz-Adame, Marica Cassarino, Elayne Ahern, Íde O'Shaughnessy, Gerard McCarthy, Rose Galvin.

**Data curation:** Dominic Trépel, Manuel Ruiz-Adame, Marica Cassarino, Collette Devlin, Rose Galvin.

**Formal analysis:** Dominic Trépel, Manuel Ruiz-Adame, Collette Devlin.

**Funding acquisition:** Marica Cassarino, Katie Robinson, Gerard McCarthy, Rose Galvin.

**Investigation:** Dominic Trépel, Manuel Ruiz-Adame, Marica Cassarino, Elayne Ahern, Katie Robinson, Íde O'Shaughnessy, Gerard McCarthy, Rose Galvin.

**Methodology:** Dominic Trépel, Manuel Ruiz-Adame, Marica Cassarino, Collette Devlin, Rose Galvin.

**Project administration:** Dominic Trépel, Manuel Ruiz-Adame.

**Resources:** Dominic Trépel, Manuel Ruiz-Adame, Marica Cassarino, Rose Galvin.

**Supervision:** Dominic Trépel, Manuel Ruiz-Adame, Marica Cassarino, Collette Devlin, Rose Galvin.

**Validation:** Dominic Trépel, Manuel Ruiz-Adame, Marica Cassarino, Elayne Ahern, Collette Devlin, Katie Robinson, Íde O'Shaughnessy, Gerard McCarthy, Cian Corcoran, Rose Galvin.

**Visualization:** Dominic Trépel, Manuel Ruiz-Adame, Marica Cassarino, Elayne Ahern, Collette Devlin, Katie Robinson, Íde O'Shaughnessy, Gerard McCarthy, Rose Galvin.

**Writing – original draft:** Dominic Trépel, Manuel Ruiz-Adame, Marica Cassarino, Rose Galvin.

**Writing – review & editing:** Dominic Trépel, Manuel Ruiz-Adame, Marica Cassarino, Elayne Ahern, Katie Robinson, Íde O'Shaughnessy, Gerard McCarthy, Rose Galvin.

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
