## [Decision Letter · Decision Letter 0]

9 Jul 2023

PONE-D-23-13587The Cost Effectiveness of Early Assessment and Intervention by a Dedicated Health and Social Care Professional Team for Older Adults in the Emergency Department Compared to Treatment-As-Usual: Economic Evaluation of the Opti-Mend TrialPLOS ONE

Dear Dr. Trépel,

Thank you for submitting your manuscript to PLOS ONE. After careful consideration, we feel that it has merit but does not fully meet PLOS ONE’s publication criteria as it currently stands. Therefore, we invite you to submit a revised version of the manuscript that addresses the points raised during the review process.

This study aims to evaluate and compare whether augmenting the treatment as usual for older adults admitted to ED is cost-effective based on the Cost-effectiveness analysis (CEA). The CEA is sound, though the use of statistical methods and regression must be justified. In addition, it is better to attach the protocol in appendix if any.

We look forward to receiving your revised manuscript.

Kind regards,

Arkers Kwan Ching Wong, Ph.D.

Academic Editor

PLOS ONE

Journal Requirements:

2. Please ensure that you refer to Figure 2 in your text as, if accepted, production will need this reference to link the reader to the figure.

3. We note you have included a table to which you do not refer in the text of your manuscript. Please ensure that you refer to Table 5 in your text; if accepted, production will need this reference to link the reader to the Table.

Additional Editor Comments:

This study aims to evaluate and compare whether augmenting the treatment as usual for older adults admitted to ED is cost-effective based on the Cost-effectiveness analysis (CEA). The CEA is sound, though the use of statistical methods and regression must be justified. In addition, it is better to attach the protocol in appendix if any.

Reviewers' comments:

Reviewer's Responses to Questions

**Comments to the Author**

1. Is the manuscript technically sound, and do the data support the conclusions?

Reviewer #1: Yes

Reviewer #2: Yes

Reviewer #3: Yes

Reviewer #4: Yes

2. Has the statistical analysis been performed appropriately and rigorously? 

Reviewer #1: Yes

Reviewer #2: Yes

Reviewer #3: I Don't Know

Reviewer #4: Yes

3. Have the authors made all data underlying the findings in their manuscript fully available?

Reviewer #1: Yes

Reviewer #2: Yes

Reviewer #3: Yes

Reviewer #4: Yes

4. Is the manuscript presented in an intelligible fashion and written in standard English?

Reviewer #1: Yes

Reviewer #2: Yes

Reviewer #3: Yes

Reviewer #4: Yes

5. Review Comments to the Author

Reviewer #1: The manuscript addresses an interesting topic. The use of QALYs is sound and in line with the existing literature. Some parts deserve a more detailed description to ensure the reproducibility and reliability of the results. Comments follow.

1. I really appreciate that the data were publicly available. This is a plus of the work. Nevertheless, I suggest to provide a more detailed description of the data and of the distribution of the outcome. The aim is to make clear which model should be used for the data at hand.

2. The modelling is briefly discussed. I strongly suggest to spend more space to the description of the statistical modelling, to the assumptions required to be met to ensure the reliability of the results. Overall, the use of a specific statistical model is not justified and not related to the analysed data. Please, ensure that the model fit is reasonable (the residual analysis is mandatory); a poor fit may lead to unreliable cost-effectiveness analysis.

3. Seemingly unrelated regression must be justified as more advanced approaches have been proposed since Zellner's seminal paper. I see that they are widely used by practitioners, mainly because they are readily available in standard softwares, but it is rather unclear why they are the "best" approach to model the data. Please, check for model's assumptions.

Reviewer #2: The authors have presented the results of a study examining the cost-effectiveness of using a team of allied health professionals to provide assesment and early intervention among older adults presenting to emergency. The methods and results are appropriate and clearly described. The resutls of this study provide a clear economic argument in favour of this intervention. My only minor comment is that the sources listed in Table 2 do not have matching entries in the Reference list at the end of the paper. Could the authors please correct this? Also, the economic protocol is unpublished. It would be helpful if this was included as a separate appendix.

Reviewer #3: It is great to see economic evaluations of this kind being carried out. Some very minor comments that OPTIMEND used in abstract vs OPTI-MEND elsewhere. Also methods section 'Resource use and associated costs' at end 'allied health service' used rather than health and social care professional service used elsewhere.

Reviewer #4: Difference in costs mainly comes from the hospital stay at baseline. It should be the effect of the intervention itself, considering the service use in other periods. Fundamentally, the intervention aims to manage patients at ED and provide appropriate assessments to reduce the useless or unhelpful hospital admission. Following up the patients beyond the index ED visit is planned to confirm that the intervention is effective and helpful, resulting in no harm to patients with the intervention. This study adds to the literature by providing the magnitude of costs and quality adjusted life years.

It would be much helpful to organize the discussion to fully explain these points. You could compare the results from studies in other settings or discuss the meaning of QALY gains (.053) in real term, etc.

6. PLOS authors have the option to publish the peer review history of their article (what does this mean?). If published, this will include your full peer review and any attached files.

Reviewer #1: No

Reviewer #2: **Yes: **Yong Yi Lee

Reviewer #3: **Yes: **Marie E. Ward

Reviewer #4: **Yes: **Deokhee Yi

---

## [Author Response · Author response to Decision Letter 0]

9 Jan 2024

To ensure that all comments were thoroughly addressed, and for ease of access for reviewers and editors of how these were have been addressed, please see my notes below (ie DT NOTES).

Journal Requirements:

• DT NOTES: The above style requirements have been checked and adjusted in line with each point

2. Please ensure that you refer to Figure 2 in your text as, if accepted, production will need this reference to link the reader to the figure.

• DT NOTES: 

o “Figure 2” had been referred to in the original article. However, it was noted this text has not been inserted as a cross-reference so did not hyperlink to figure – this has now been updated.

3. We note you have included a table to which you do not refer in the text of your manuscript. Please ensure that you refer to Table 5 in your text; if accepted, production will need this reference to link the reader to the Table.

• DT NOTES: 

o The paragraph relating to results of seemingly unrelated regression now concludes to point the read to the table of results by saying “(see Table 5).”

4. Please review your reference list to ensure that it is complete and correct. If you have cited papers that have been retracted, please include the rationale for doing so in the manuscript text, or remove these references and replace them with relevant current references. 

• DT NOTES: 

o Reference list from 1 to 15 are all complete and correct and I do not find any need to retract or remove any references.

o I have noted an issue with reference manager from my former researcher of different software and have addressed this by ensuring all citation were included using Endnote. All former references were removed and replacedin the required style

o I note that reference to the primary paper (ie reference 3) was cited multiple in the body text and therefore are superfluous to need and are therefore retracted. The one exception that is kept is the citation in Figure 1 as it is felt this is vital to clarifying differences between complete case analyses between the clinical and cost effectiveness papers

o I also note that reference to Zellner 1962 paper on ‘seemingly unrelated regression’ is missing and is now added (appears now at reference 9 making a total of 16 references). This omission also addresses query by one reviewer (see later comment regarding this reviewer comment). 

o Also, references to sources of costs (Table 2) were missing and now figure as reference 18 (Gillespie 2022) and 19 (Smith 2021).

o Finally, in line with PLOS instructions on referencing style, Endnote style was set to ‘Vancouver’. However, contrary Plos “MANUSCRIPT BODY FORMATTING GUIDELINES” where is state “Cite references in brackets (for example, “[1]” or “[2-5]”or “[3,7,9])”, please note Vancouver appears with round brackets. As such, Endnote template for Vancouver has been adjusted (see link).

Any changes to the reference list should be mentioned in the rebuttal letter that accompanies your revised manuscript. If you need to cite a retracted article, indicate the article’s retracted status in the References list and also include a citation and full reference for the retraction notice.

• DT NOTES:

o Changes to refences are given in detail here and referenced in the letter. The citation list is now 100% accurate

Additional Editor Comments:

This study aims to evaluate and compare whether augmenting the treatment as usual for older adults admitted to ED is cost-effective based on the Cost-effectiveness analysis (CEA). The CEA is sound, though the use of statistical methods and regression must be justified. 

• DT NOTES: 

o The comment on statistical methods (Zellner 1962 seemingly unrelated requestions) was missing. 

o In relation to the reviewer comment (ie “Seemingly unrelated regression must be justified as more advanced approaches have been proposed since Zellner's seminal paper”), I further add an explanatory text 

o 

o … article of the statistical approach in Health Economics (see Willan 2004, citation 16) and more recent article entitled ‘The statistical approach in trial-based economic evaluations matters: get your statistics together” (see citation 17)

o Finally, for complete clarify, under results I emphasise how the correlation from the residual matrix can be interpreted by adding “Correlation between Total Cost and QALYs was -0.2803 and negative correlation indicates individuals with worse outcomes have higher costs.”

In addition, it is better to attach the protocol in appendix if any.

• DT NOTES: 

o There is a published trial protocol which is referenced (see citation number 12)

o Furthermore, as per requirements (see CHEERS checklist) a Health Economic Analysis Plan was also peer reviewed and published (see citation 11).

---

## [Editor Report · Decision Letter 1]

22 Jan 2024

The Cost Effectiveness of Early Assessment and Intervention by a Dedicated Health and Social Care Professional Team for Older Adults in the Emergency Department Compared to Treatment-As-Usual: Economic Evaluation of the Opti-Mend Trial

PONE-D-23-13587R1

Dear Dr. Trépel,

We’re pleased to inform you that your manuscript has been judged scientifically suitable for publication and will be formally accepted for publication once it meets all outstanding technical requirements.

Kind regards,

Arkers Kwan Ching Wong, Ph.D.

Academic Editor

PLOS ONE
---

## [Editor Report · Acceptance letter]

13 Jun 2024

PONE-D-23-13587R1 

PLOS ONE

Dear Dr. Trépel, 

I'm pleased to inform you that your manuscript has been deemed suitable for publication in PLOS ONE. Congratulations! Your manuscript is now being handed over to our production team.

Kind regards, 

on behalf of

Dr Arkers Kwan Ching Wong 

Academic Editor

PLOS ONE